# Shared Genomic Features Between Lung Adenocarcinoma and Type 2 Diabetes: A Bioinformatics Study

**DOI:** 10.3390/biology14040331

**Published:** 2025-03-25

**Authors:** Nuerbiye Nueraihemaiti, Dilihuma Dilimulati, Alhar Baishan, Sendaer Hailati, Nulibiya Maihemuti, Alifeiye Aikebaier, Yipaerguli Paerhati, Wenting Zhou

**Affiliations:** 1Department of Pharmacology, School of Pharmacy, Xinjiang Medical University, Urumqi 830017, China; nurbiye@stu.xjmu.edu.cn (N.N.); dilihuma@stu.xjmu.edu.cn (D.D.); alhar@stu.xjmu.edu.cn (A.B.); sendaer@stu.xjmu.edu.cn (S.H.); nurbiye24@stu.xjmu.edu.cn (N.M.); alifeiye@stu.xjmu.edu.cn (A.A.); yipaerguli@stu.xjmu.edu.cn (Y.P.); 2Xinjiang Key Laboratory of Natural Medicines Active Components and Drug Release Technology, Urumqi 830017, China; 3Xinjiang Key Laboratory of Biopharmaceuticals and Medical Devices, Urumqi 830017, China; 4Engineering Research Center of Xinjiang and Central Asian Medicine Resources, Ministry of Education, Urumqi 830017, China

**Keywords:** lung adenocarcinoma (LUAD), type 2 diabetes (T2DM), bioinformatics, genomic features

## Abstract

Lung adenocarcinoma, a prevalent histological variant of lung cancer, is experiencing a persistent rise in global incidence. Concurrently, type 2 diabetes presents a significant risk to human health. However, the molecular mechanisms connecting these two diseases remain predominantly unexplored. This study used bioinformatics analysis techniques. Our objective was to identify the potential genes linked to type 2 diabetes and lung adenocarcinoma, as well as to investigate the associated microRNAs (miRNAs) and transcription factor (TF) genes. The research findings indicate that ATR, RFC4, and MCM2 are pivotal genes in the pathogenesis of type 2 diabetes and lung adenocarcinoma. Moreover, hsa-mir147a, hsa-mir16-5p, and hsa-mir-1-3p exhibit the most significant correlation with these hub genes. These findings enhance the comprehension of the pathogenesis of type 2 diabetes and lung adenocarcinoma while offering critical insights for refining management plans and augmenting treatment strategies for both conditions.

## 1. Introduction

Lung cancer is among the most common malignancies and continues to be a primary cause of cancer-related mortality globally [1]. The incidence and mortality rates have been consistently rising [2], with an estimated 20,000 new cases and 20,000 deaths each year [3]. The five-year survival rate for lung cancer varies internationally, ranging from 10% to 20% in most nations, with Japan exhibiting the highest rate at 33% [4]. Lung cancer is categorized into small cell lung cancer (SCLC) and non-small cell lung cancer (NSCLC), with NSCLC representing roughly 80% of cases [5]. Lung adenocarcinoma (LUAD) is the predominant histological variant among NSCLC subtypes. Although early-stage LUAD patients may benefit from surgical resection, patients in advanced stages necessitate alternative therapies. The advent of molecular-targeted therapies and immune checkpoint inhibitors has considerably broadened treatment alternatives for LUAD [6,7]. Comprehending the molecular mechanisms that govern alterations in gene expression in LUAD is essential for enhancing patient survival rates.

Diabetes mellitus (DM) is a chronic metabolic condition marked by consistently elevated blood glucose levels. The two main subtypes are type 1 diabetes mellitus (T1DM) and type 2 diabetes mellitus (T2DM), each characterized by unique pathophysiological mechanisms [8]. T1DM arises from the autoimmune destruction of pancreatic β-cells, resulting in complete insulin deficiency and a lifelong requirement for insulin therapy. Conversely, T2DM is marked by insulin resistance in peripheral tissues, including the liver, muscles, and adipose tissue, alongside progressive dysfunction of pancreatic β-cells, ultimately leading to insufficient insulin secretion. Diabetes is linked to microvascular complications, including nephropathy [9] and retinopathy [10], but also with an increased risk of cancers, including lung, breast, and pancreatic cancer [11], breast cancer [12], and pancreatic cancer [13]. Yang et al. examined clinical data from hospitalized T2DM patients diagnosed with diverse cancers over a five-year period and determined that lung cancer, specifically LUAD, was the predominant malignancy among this cohort [14].

Lai et al. additionally reported that T2DM is associated with reduced serum 25-hydroxyvitamin D3 levels, with diminished levels correlating with an elevated risk of LUAD [15]. Guo et al. discovered that in LUAD, the long non-coding RNA metastasis-associated lung adenocarcinoma transcript 1 modulates downstream functional proteins, affecting the progression of diabetes and its complications [16]. This lncRNA competitively binds to microRNAs (miRNAs), influencing downstream transcription factors (TFs) associated with inflammation and apoptosis, which may contribute to the progression of LUAD and diabetes-related complications. These findings indicate a potential bidirectional relationship between LUAD and T2DM, wherein diabetes may facilitate cancer initiation and progression. Nonetheless, the exact mechanisms underlying this association remain ambiguous.

This study aimed to investigate the shared molecular mechanisms between LUAD and T2DM. We acquired datasets (GSE40791, GSE25724, GSE10072, and GSE71416) from the Gene Expression Omnibus (GEO) database, encompassing samples from LUAD patients, T2DM patients, and healthy controls. Differentially expressed genes (DEGs) were identified, and the common DEGs between LUAD and T2DM were subjected to further analysis. Considering the possible association between these diseases and immune dysregulation, we conducted single-sample gene set enrichment analysis (ssGSEA) to evaluate immune infiltration patterns in both LUAD and T2DM. Furthermore, gene ontology (GO) and Kyoto Encyclopedia of Genes and Genomes (KEGG) pathway enrichment analyses were performed to clarify the biological functions and pathways related to the common DEGs. A protein–protein interaction (PPI) network was established to identify hub genes, along with their corresponding miRNAs and transcription factors (TFs). Our findings elucidate the common molecular mechanisms linking LUAD and T2DM, presenting potential targets for forthcoming diagnostic and therapeutic approaches.

## 2. Materials and Methods

### 2.1. Data Collection

We established the following as our screening criteria: ensuring data loss in either the samples or the probes, the data align with the research objectives, and in the experimental design, there are scientifically sound and reasonable blank and positive controls. Based on these criteria, we precisely searched for datasets related to “lung adenocarcinoma” and “diabetes” in the Gene Expression Omnibus (GEO, https://www.ncbi.nlm.nih.gov/geo/ (accessed on 5 January 2024)) of the National Center for Biotechnology Information (NCBI). After a series of meticulous screenings, we finally identified four datasets, namely GSE40791 (LUAD), GSE25724 (T2DM), GSE10072 (LUAD), and GSE71416 (T2DM), for further in-depth analysis. The sequencing platform used for GSE40791 (LUAD) and GSE71416 (T2DM) was GPL570, whereas GPL96 was used for GSE25724 (T2DM) and GSE10072 (LUAD).

GSE40791 (LUAD) dataset: This dataset consists of 194 samples, with 100 samples from non-lung adenocarcinoma cases and 94 samples from lung adenocarcinoma cases.

GSE25724 (T2DM) dataset: Contains 13 samples, among which 7 are non-type 2 diabetes samples and 6 are type 2 diabetes samples.

GSE10072 (LUAD) dataset: There are a total of 107 samples, with 49 samples from non-lung adenocarcinoma cases and 58 samples from lung adenocarcinoma cases.

GSE71416 (T2DM) dataset: Includes 20 samples, 6 from non-diabetes samples and 14 from diabetes samples. The GSE71416 (T2DM) and GSE10072 (LUAD) datasets respectively serve as external validation datasets for the corresponding diseases. By performing validation analyses on this dataset, we assessed the stability and consistency of key genes across different datasets and sample conditions, thereby enhancing the reliability of the study results.

### 2.2. Identification of DEGs

The R packages “limma” and “GEOquery” (version 4.1.3) were used for data normalization and probe annotation of the GSE40791 and GSE25724 datasets. DEGs were identified using an adjusted *p*-value threshold of <0.05 and an absolute log2 fold change > 1.5. The R package “ggplot” was used to generate volcano plots for visualizing the DEGs in the datasets. The heat map was generated using the “heatmap” function in R. Venn diagram analysis was conducted using the online tool (http://www.liuxiaoyuyuan.cn/ (accessed on 7 January 2024)) to extract the DEGs commonly observed between GSE40791 and GSE25724.

### 2.3. Immune Infiltration Analysis

The ssGSEA (http://www.biocloudservice.com/home.html (accessed on 10 February 2024)) score was used to quantify immune cell infiltration in LUAD and T2DM tissues, providing an infiltration level for each dataset sample. The ssGSEA score was used to quantify the infiltration of immune cells in LUAD or T2DM tissues and determine the level of immune infiltration in each dataset sample. Spearman correlation analysis was conducted to determine the correlation between DEGs and 28 immune cells.

### 2.4. Enrichment and Pathway Analysis

Using the DAVID online tool (https://davidbioinformatics.nih.gov/ (accessed on 13 February 2024)), we performed gene ontology (GO) annotation and Kyoto Encyclopedia of Genes and Genomes (KEGG) pathway enrichment analysis on the common DEGs to elucidate their functions. The enriched GO terms and KEGG pathways with a *p*-value < 0.05 were analyzed and visually presented using both bubble and bar graphs.

### 2.5. Construction of Protein–Protein Interaction (PPI) Network

We employed the protein interaction gene search tool STRING (http://string-db.org/ (accessed on 15 February 2024)) for the analysis of PPI of the common DEGs. We extracted PPI pairs with interaction scores above 0.9 and visualized them using Cytoscape 3.9.1 (https://cytoscape.org/release_notes_3_9_1.html (accessed on 17 February 2024)). Furthermore, we used the Cytoscape plugin MCODE to identify significant functional modules, with the following selection criteria: K-core of 2, degree cutoff of 2, maximum depth of 100, and node score cutoff of 0.2.

### 2.6. Identification of Hub Genes

The CytoHubba (https://cytoscape.org/release_notes_3_9_1.html (accessed on 20 February 2024)) plugin of Cytoscape was used to identify hub genes from the PPI network, applying four algorithms—MNC, MCC, EPC, and Degree—to confirm the key hub genes. A Venn diagram was used for visualization.

### 2.7. Receiver Operating Characteristic (ROC) Curve Analysis of Hub Genes

The pROC software package (version 4.1.3) in R language was used to generate ROC curves to evaluate the predictive values of hub genes in two test datasets, including GSE10072 (LUAD) and GSE71416 (T2DM).

### 2.8. Prediction of miRNAs for the Central Hub Genes

To gain a deeper understanding of the relationship between miRNAs and target genes of T2DM and LUAD, we used the NetworkAnalyst dataset (https://www.networkanalyst.ca/ (accessed on 5 March 2024)) to predict the miRNAs of the central hub genes. The KEGG enrichment analysis of three common miRNAs (hsa-mir147a, hsa-mir16-5p, and hsa-mir-1-3p) was conducted using miRPath4.0 (http://www.microrna.gr/miRPathv4 (accessed on 8 March 2024)) to identify signaling pathways.

### 2.9. Construction of Hub Gene-TF Regulatory Network

To investigate the interactions between TFs and central hub genes and evaluate the influence of TFs on the expression and functional pathways of central hub genes, we employed NetworkAnalyst. We predicted the TFs for central hub genes from this database and visualized the transcriptional regulatory network using the Cytoscape software.

## 3. Results

### 3.1. Identification of Common DEGs Between T2DM and LUAD

After screening based on the criteria of *p*-value < 0.05 and |log2 fold change| > 1.5, a total of 4598 DEGs from GSE40791 and 2790 DEGs from GSE25724 were identified, which are presented in volcano plots (Figure 1A,B) and heat maps (Figure 1C,D), respectively. Furthermore, using Venn diagram analysis, 748 DEGs were identified at the intersection of LUAD DEGs and TDM DEGs, after excluding genes with opposite expression trends (Figure 1E). There were 99 upregulated DEGs and 526 downregulated DEGs.

### 3.2. Immune Infiltration Analysis in LUAD and T2DM

To investigate the immunocyte infiltration in the control and case groups of LUAD and T2DM, we performed the ssGSEA analysis. Figure 2A demonstrates the correlation between LUAD and immune cells, including Central memory CD8 T cell, Effector memory CD8 T cell, Activated CD4 T cell, Effector memory CD4 T cell, T follicular helper cell, Type 1 T helper cell, Type 17 T helper cell, Type 2 T helper cell, Regulatory T cell, Activated B cell, Memory B cell, Natural killer cell, CD56bright natural killer cell, CD56dim natural killer cell, Myeloid-derived suppressor cell, Natural killer T cell, Activated dendritic cell, Plasmacytoid dendritic cell, Immature dendritic cell, Macrophage, Eosinophil, Mast cell, Monocyte, and Neutrophil. As shown in Figure 2B, T2DM was correlated with immune cells, including the Effector memory CD8 T cell, Effector memory CD4 T cell, T follicular helper cell, Type 1 T helper cell, Activated B cell, Natural killer cell, Natural killer T cell, Immature dendritic cell, Macrophage, and Neutrophil.

### 3.3. Enrichment Analysis of Common DEGs

To decipher the biological functions, GO (Figure 3A) and KEGG (Figure 3B) functional enrichment analyses were performed using the DAVID online tool on 748 common DEGs of LUAD and T2DM. These genes were primarily involved in biological processes such as the positive regulation of I-κB kinase/NF-κB signaling, positive regulation of immunoglobulin production, positive regulation of cell proliferation, cellular response to interleukin-7, cellular response to interleukin-4, and positive regulation of TOR signaling. Molecular functions of these genes included protein binding, lipid binding, protein homodimerization activity, and ubiquitin protein ligase binding, among others. In terms of cellular components, these genes were predominantly associated with the cytosol, extracellular vesicles, endoplasmic reticulum membrane, and cytoplasm. Additionally, KEGG pathway analysis revealed involvement in the transforming growth factor-beta (TGF-β) signaling pathway, C-type lectin receptor signaling pathway, lipid metabolism, and atherosclerosis.

### 3.4. PPI Network

To investigate the interactions between DEGs, we constructed a PPI network using the STRING database. A visual interaction score of 0.9 was set as the minimum threshold for constructing the PPI network. This PPI network consisted of 739 nodes and 577 edges, with a PPI enrichment *p*-value < 2.75 × 10^−5^ (Figure 4A). Using the MCODE plugin in Cytoscape, we identified three significant modules (Figure 4B–D) comprising 11 nodes and 56 edges, 10 nodes, and 46 edges, and 9 nodes and 34 edges, respectively. By employing four algorithms (MCC, MNC, Degree, and EPC) in the CytoHubba plugin, we obtained the top 20 hub genes and visualized the overlapping central hub genes among these algorithms using a Venn diagram. Finally, seven central hub genes were identified, including *MCM2*, *RFC4*, *ATR*, *NUP155*, *NUP107*, *NUP85*, and *NUP37* (Figure 4E).

### 3.5. ROC Analysis of Hub Genes

We used the pROC package in R to plot the ROC curve for the hub genes and evaluate their predictive accuracy. Our training set included GSE40791 and GSE25724, while the validation set consisted of GSE10072 (LUAD) and GSE71416 (T2DM). In all the datasets, including GSE10072 and GSE71416, the area under the curve values of the three hub genes, *RFC4*, *MCM2*, and *ATR*, were all greater than 0.7, indicating their high predictive reliability (Figure 5). Therefore, we selected these three hub genes as the central hub genes.

### 3.6. Network of Central Hub Genes and miRNAs

To gain a deeper understanding of the relationship between miRNAs and target genes of T2DM and LUAD, we constructed and utilized the miRNA–target gene network using the NetworkAnalyst dataset. We displayed the central hub genes and the associated regulatory miRNAs in Figure 6. The results revealed that the degree of hsa-mir147a, hsa-mir16-5p, and hsa-mir-1-3p were greater than 3, indicating a stronger association with T2DM and LUAD. Therefore, these candidate miRNAs may provide a solid foundation for understanding the molecular mechanisms of T2DM and LUAD, as well as revealing a series of promising shared targets in T2DM and LUAD.

### 3.7. KEGG Pathway Enrichment Analysis of hsa-mir147a, hsa-mir16-5p, and hsa-mir-1-3p

We performed KEGG enrichment analysis on hsa-mir147a, hsa-mir16-5p, and hsa-mir-1-3p using miRPath4.0. The results demonstrated that multiple signaling pathways were identified, including the TGF-β signaling pathway, Rap1 signaling pathway, MAPK signaling pathway, IL-17 signaling pathway, PI3K-Akt signaling pathway, and other signaling pathways (Table 1). Therefore, these three miRNAs may play a role in the dysregulation of signal transduction and immune system pathways in T2DM and LUAD.

### 3.8. Construction of TF Regulatory Network

Identifying common TFs will help us understand the possible mechanisms related to T2DM and LUAD. We used NetworkAnalyst 3.0 to analyze the interactions between the central hub genes and TFs and evaluate the impact of TFs on the expression and functional pathways of the central hub genes. A hub gene–TF regulatory network consisting of 67 TFs was constructed (Figure 7). *ATR* was regulated by 13 TFs, *MCM2* was regulated by 20 TFs, and *RFC4* was regulated by 34 TFs. Additionally, in the hub gene-TF regulatory network, both specificity protein 1 (SP1) and lysine-specific demethylase 5A (KDM5A) regulated the central hub genes. Therefore, these two regulatory factors were identified as key TFs that were crucial for guiding gene expression and the physiological and pathological processes of T2DM and LUAD.

## 4. Discussion

Diabetes mellitus (DM) is a chronic metabolic condition with substantial genetic foundations, although its exact etiology and pathogenesis are still not fully understood. As of now, roughly 463 million adults between the ages of 20 and 79 are afflicted by diabetes, constituting 9.3% of the global population within this demographic [17]. Although diabetes is typically linked to microvascular complications, there is growing evidence connecting it to multiple cancers. Lung adenocarcinoma (LUAD) has been recognized as a significant complication linked to extended diabetes duration [15,18]. Research has established correlations between type 2 diabetes mellitus (T2DM) and various malignancies, including cancer [19], breast cancer [20], endometrial cancer [21], hepatocellular carcinoma [22], pancreatic cancer [23], lung cancer [24], renal cell carcinoma [25], and cervical cancer [26,27]. Considering that genetic factors significantly affect T2DM susceptibility, it is posited that diabetes may not only coexist with but also facilitate cancer development [28]. However, the specific molecular mechanisms connecting diabetes and cancer remain predominantly unexamined. Consequently, our research examines the interaction between T2DM and LUAD.

We conducted a bioinformatics analysis using datasets GSE24724 and GSE40791 from the GEO database to explore the common molecular mechanisms linking T2DM and LUAD. A total of 738 DEGs exhibiting analogous expression patterns in T2DM and LUAD were identified and analyzed for functional enrichment. GO and KEGG pathway analyses indicated that these DEGs were primarily enriched in immune-related pathways, encompassing the positive regulation of I-κB kinase/NF-κB signaling, enhancement of immunoglobulin production, cellular responses to interleukin-7 and interleukin-4, positive regulation of the TOR signaling pathway, the TGF-β signaling pathway, and the C-type lectin receptor signaling pathway. These findings underscore the pivotal role of inflammation in the pathogenesis of both type 2 diabetes mellitus and lung adenocarcinoma. The TGF-β signaling pathway is recognized for its regulation of cell proliferation, differentiation, apoptosis, migration, and tumor progression [29,30]. In early-stage cancer, TGF-β demonstrates tumor-suppressive effects by inducing G1 cell cycle arrest and apoptosis [31]. Subsequently, TGF-β facilitates EMT, stemness, metastasis, drug resistance, and malignant progression [32]. Elevated glucose levels in T2DM have been demonstrated to increase pro-inflammatory cytokines, such as TNF-α, IFN-γ, and TGF-β, which may augment TGF-β pathway activation and heighten the risk of LUAD [33].

Analysis of immune infiltration demonstrated substantial correlations among LUAD, T2DM, and various immune cell populations, including effector memory CD8+ T cells, T follicular helper cells, type 1 T helper cells, activated B cells, NK cells, macrophages, and neutrophils. NK cells, comprising 5% to 19% of peripheral blood lymphocytes, are crucial in anti-tumor immunity via the secretion of cytokines and chemokines [34,35]. Decreased NK cell activity has been associated with heightened cancer susceptibility [36], and modified NK cell function has been noted in obesity and T2DM [37,38]. In type 2 diabetes mellitus, natural killer cells demonstrate diminished cytotoxicity and decreased population levels. Additionally, they promote TGF-β pathway activation through the secretion of TNF-α and IFN-γ, thereby advancing LUAD progression [39,40,41]. Impaired NK cell function may therefore contribute to both elevated cancer risk and increased cancer incidence in patients with T2DM [35].

*ATR* is expressed in multiple tumor types. Inhibition of *ATR* and TOP1 is reported to augment the immunogenicity of small-cell lung cancer [42]. Furthermore, research has demonstrated the elevated expression of *MCM2* in various cancers, including breast cancer [43] and hepatocellular carcinoma [44], suggesting that *MCM2* may represent a viable therapeutic target for numerous cancer types. *RFC4* has been documented in nasopharyngeal carcinoma [45], oral squamous cell carcinoma [46], and various other malignancies. *RFC4* has been identified as a potential pathogenic mechanism in T2DM [47]. We used the CytoHubba plugin of Cytoscape to identify potential hub genes. Seven genes were identified: *ATR*, *MCM2*, *RFC4*, *NUP155*, *NUP107*, *NUP85*, and *NUP37*. Subsequently, we delineated the ROC curves for these seven prospective hub genes. The findings indicated that the AUC values for *ATR*, *RFC4*, and *MCM2* exceeded 0.7 in both validation datasets. Consequently, we identified these three genes as the primary hub genes for subsequent investigation.

MicroRNAs are being extensively researched for their functions in development, diagnosis, and prognosis across various domains. A recent study indicated that miR-1 serves as a dependable predictor of myocardial lipotoxicity in T2DM [48]. Hsa-mir-1-3p is ranked third in non-small cell lung cancer relative to normal lung tissue [49]. Hsa-mir16-5p is a crucial biomarker for the diagnosis of Alzheimer’s disease [50]. Currently, there is no report associating hsa-mir16-5p with LUAD and T2DM. In our study, we developed a central hub gene–miRNA network to investigate the potential regulatory mechanisms of central hub genes in T2DM and LUAD. The findings indicated that hsa-mir147a, hsa-mir16-5p, and hsa-mir-1-3p exhibited the greatest connectivity with the three central hub genes. This signifies their significance in the possible mechanisms of LUAD and T2DM. No studies exist regarding the role of mir-147a in LUAD and T2DM. It has been reported that miR-147a modulates the progression of NSCLC by targeting CC chemokine ligand 5 [51]. Furthermore, miR-147a directly interacts with myocardial infarction-associated transcript (MIAT), and E2F transcription factor 3 (E2F3) has been established as the target gene of miR-147a [52]. Altered expression of MIAT is implicated in cellular apoptosis and the pathogenesis of several diseases, including myocardial infarction, microvascular dysfunction, diabetes, and cancer [53]. miR-147a significantly inhibits E2F3 expression, while the competitive interaction between MIAT and miR-147a liberates E2F3. Thus, the MIAT/miR-147a/E2F3 axis may promote cellular proliferation and fibrosis in the progression of diabetic nephropathy [52].

Transcription factors are crucial in modulating gene expression. In the hub gene-TF regulatory network, we noted that SP1 and KDM5A exhibited a greater interaction frequency with the three central hub genes. SP1 is a transcription factor of the Sp/KLF family that operates by binding to GC-rich sequences in gene regulatory regions. SP1 regulates cell proliferation, apoptosis, differentiation, and angiogenesis [54]. SP1 is significantly overexpressed in gastric cancer [55], meningioma [56], lung adenocarcinoma [57], liver cancer [58], and pancreatic cancer [59]. SP1 engages with the gene regulatory regions that encode components of the MAPK, p38, JAK/STAT, and PI3K/Akt signaling pathways, thereby influencing cell proliferation, differentiation, and apoptosis [60]. KDM5A/RBP2 is a demethylase that suppresses NOTCH signaling, sustains neuroendocrine differentiation, and facilitates the development of small-cell lung cancer [61].

T2DM and LUAD share a complex bidirectional relationship that significantly influences patient diagnosis, prognosis, and treatment. The fundamental molecular mechanisms likely entail complex and interconnected interactions among various signaling pathways that are essential in the pathogenesis of both diseases. This study identifies *ATR*, *RFC4*, and *MCM2* as pivotal crosstalk genes that significantly influence T2DM and LUAD. *ATR* is essential for preserving the genomic stability of pancreatic islet β-cells. After *ATR* malfunction, insulin secretion will be compromised. *MCM2* participates in the regulation of the cell cycle and DNA replication. Aberrant *MCM2* function can interfere with normal cellular metabolism, significantly impacting the pathogenesis of T2DM. If *ATR* undergoes mutations or exhibits abnormal functionality, it will result in the disruption of the regulatory mechanisms governing cell proliferation, thereby posing a latent risk for the development of lung adenocarcinoma. *RFC4* is crucial in the DNA repair mechanism. When *RFC4* malfunctions, the cell’s capacity to repair DNA damage is markedly diminished, thereby hastening the onset and progression of LUAD. Aberrant *MCM2* function disrupts the normal cell cycle and exacerbates the progression of LUAD. These pivotal crosstalk genes create intricate molecular linkages between the two diseases, offering a novel perspective and theoretical foundation for a more profound comprehension of the pathogenesis of T2DM and LUAD, as well as the formulation of more efficacious diagnostic and therapeutic strategies.

## 5. Conclusions

This bioinformatics study identified common genetic traits between T2DM and LUAD, elucidating potential molecular mechanisms involved in their pathogenesis. Our analysis indicates that dysregulated immune responses and abnormal TGF-β signaling may serve as shared pathogenic mechanisms connecting LUAD and T2DM. Moreover, *ATR*, *MCM2*, and *RFC4* have emerged as prospective therapeutic targets for both conditions, necessitating further exploration. Our findings establish a basis for future research focused on creating innovative therapeutic and preventive approaches for T2DM and LUAD.

## Figures and Tables

**Figure 1 biology-14-00331-f001:**
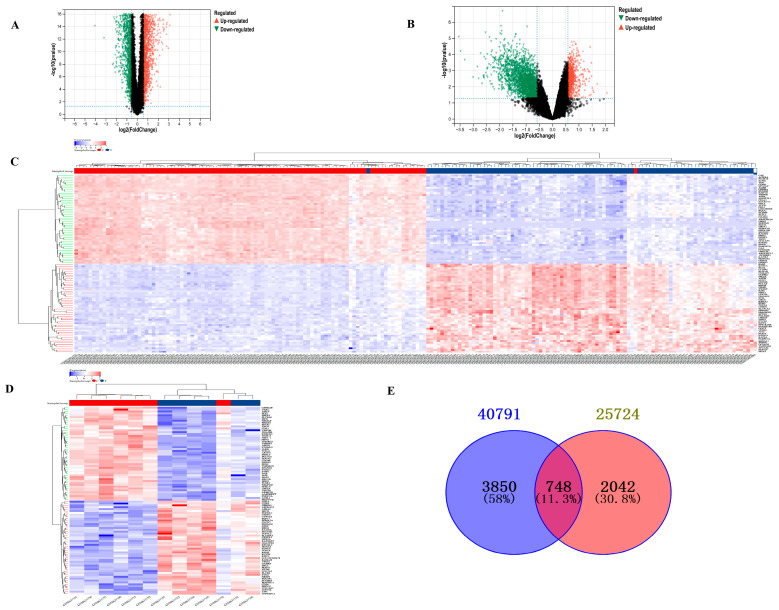
Analysis of DEGs between T2DM and LUAD. (**A**) Volcano plot showing DEGs of LUAD from GSE40791. (**B**) Volcano plot showing DEGs of T2DM from GSE25724. (**C**) Heat map showing DEGs of T2DM from GSE40791. (**D**) Heat map showing DEGs of LUAD from GSE25724. (**E**) Venn diagram showing common DEGs between T2DM and LUAD.

**Figure 2 biology-14-00331-f002:**
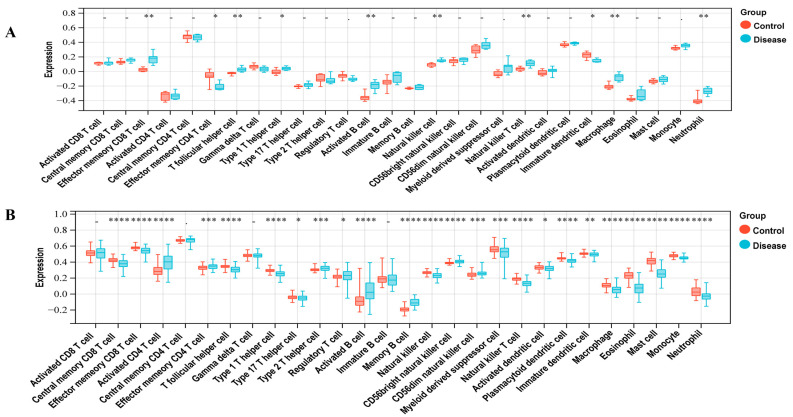
Immune infiltration analysis. (**A**) Correlation between LUAD with immune cells. (**B**) Correlation between T2DM with immune cells. * represents a *p*-value < 0.05; ** represents a *p*-value < 0.01; *** represents a *p*-value < 0.001; **** represents a *p*-value < 0.0001.

**Figure 3 biology-14-00331-f003:**
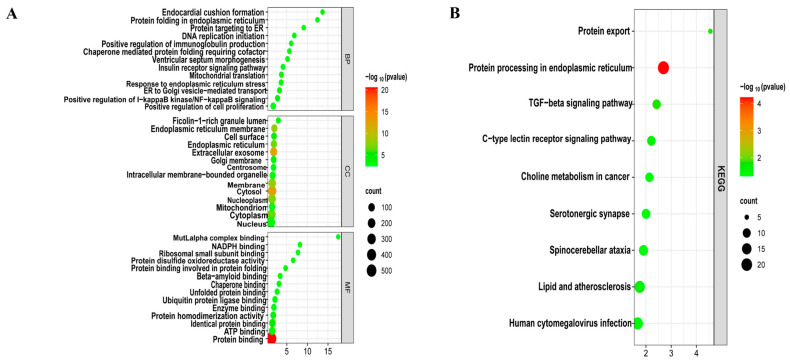
Enrichment analysis of common DEGs. (**A**) GO enrichment analysis. Top GO terms in biological process (BF), cellular component (CC), and molecular function (MF) are displayed. (**B**) KEGG pathway enrichment analysis. Top 9 pathways are presented.

**Figure 4 biology-14-00331-f004:**
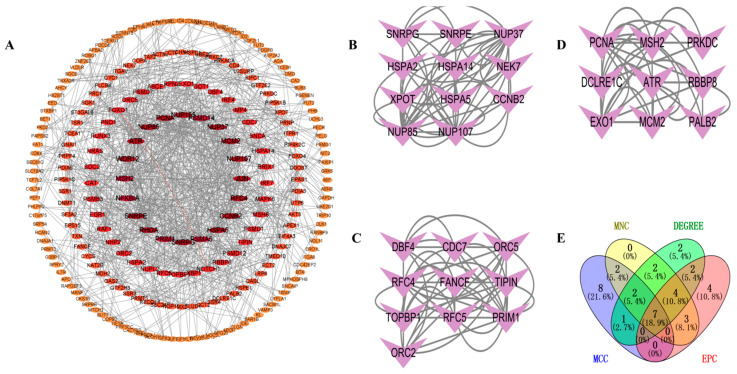
PPI network analysis. (**A**) PPI network. (**B**) Significant module 1. (**C**) Significant module 2. (**D**) Significant module 3. (**E**) Venn diagram of MNC, DEGREE, EPC, and MCC.

**Figure 5 biology-14-00331-f005:**
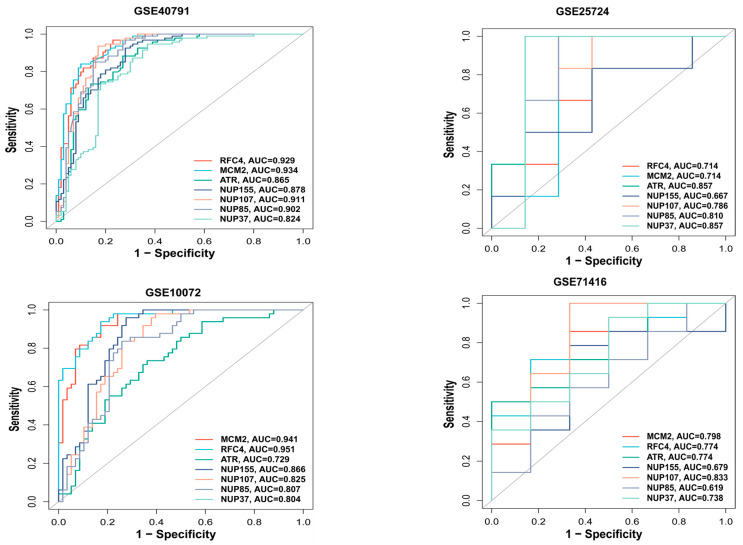
ROC analysis of hub genes. ROC curve for hub genes from the training set (GSE40791 and GSE25724) and the validation set (GSE10072 and GSE71416) were plotted with the pROC package in R.

**Figure 6 biology-14-00331-f006:**
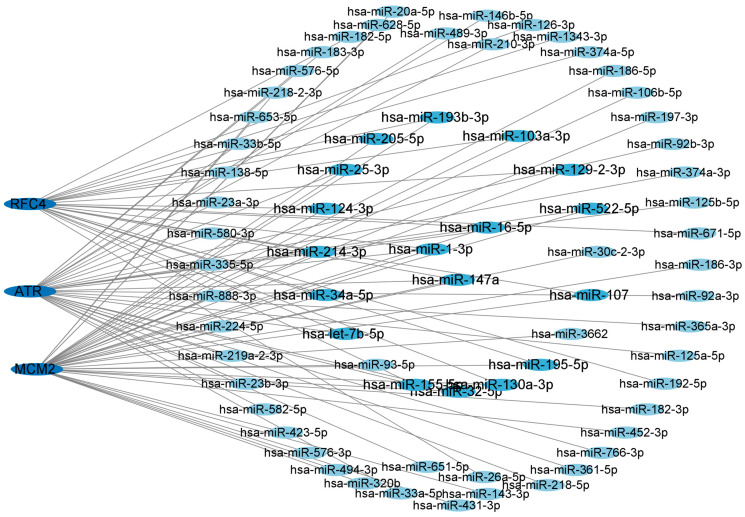
Network of central hub genes and miRNAs. miRNA–target gene network was constructed using the NetworkAnalyst dataset. Central hub genes were *RFC4*, *MCM2*, and *ATR*.

**Figure 7 biology-14-00331-f007:**
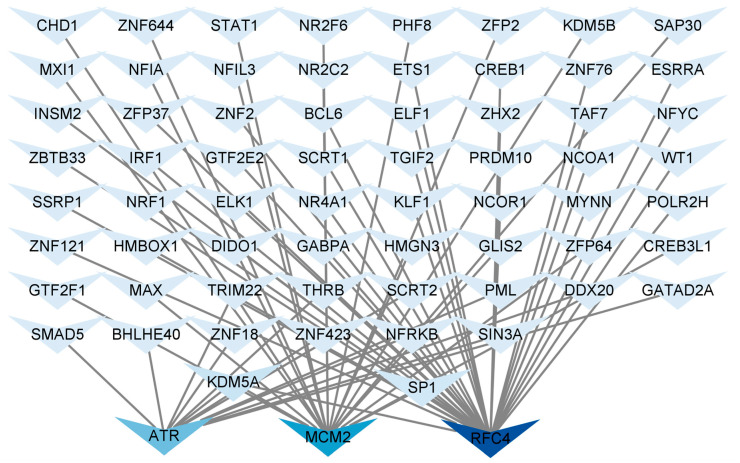
Regulatory network of central hub genes and TFs. Hub gene–TF regulatory network was constructed using the NetworkAnalyst. Central hub genes were *RFC4*, *MCM2*, and *ATR*.

**Table 1 biology-14-00331-t001:** Enriched KEGG pathway.

Term Name	Term Genes	Target Genes (n)	miRNAs (n)	miRNA Names	*p*-Value
TGF-beta signaling pathway	103	36	2	hsa-miR-1-3p, hsa-miR-16-5p	1.7046 × 10^−7^
IL-17 signaling pathway	106	24	2	hsa-miR-1-3p, hsa-miR-16-5p	0.016457779
Rap1 signaling pathway	214	59	3	hsa-miR-1-3p, hsa-miR-147a, hsa-miR-16-5p	4.37555 × 10^−7^
MAPK signaling pathway	329	75	3	hsa-miR-1-3p,hsa-miR-147a, hsa-miR-16-5p	3.20369 × 10^−5^
PI3K-Akt signaling pathway	372	76	3	hsa-miR-1-3p, hsa-miR-147a, hsa-miR-16-5p	0.001053445

## Data Availability

The datasets (GSE40791, GSE25724, GSE, GSE10072, GSE71416) for this study can be found in the Gene Expression Omnibus (GEO) database (https://www.ncbi.nlm.nih.gov/geo/ (accessed on 5 January 2024)) and the STRING database (https://cn.string-db.org/ (accessed on 15 February 2024)). All the data in this paper support the results of this study.

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
