# Peer review of "Shared Genomic Features Between Lung Adenocarcinoma and Type 2 Diabetes: A Bioinformatics Study"

_biology, 2025, doi:10.3390/biology14040331_

Round 1
Reviewer 1 Report
Comments and Suggestions for Authors
Article Summary:
The article, “Shared Genomic Features between Lung Adenocarcinoma and Type 2 Diabetes: A Bioinformatics Study”, explores the molecular connections between LUAD and T2DM using bioinformatics analyses. The study identifies 738 shared differentially expressed genes (DEGs) and highlights key immune-related pathways, including the TGF-β signaling pathway, as potential contributors to both diseases. Through network analysis, ATR, MCM2, and RFC4 are identified as central hub genes with potential therapeutic relevance. Additionally, the study investigates immune cell infiltration patterns and the role of miRNAs and transcription factors in regulating shared genetic features. The findings suggest that immune dysregulation and chronic inflammation may play a crucial role in linking LUAD and T2DM. This research provides new insights into the genetic and molecular interplay between these diseases, offering potential targets for future therapeutic and preventive strategies.
Review:
The manuscript titled "Shared Genomic Features between Lung Adenocarcinoma and Type 2 Diabetes: A Bioinformatics Study" investigates the molecular and genetic connections between LUAD and T2DM, identifying shared differentially expressed genes (DEGs) and key pathways. The study highlights the role of immune dysregulation and the TGF-β signaling pathway in both diseases, emphasizing their contribution to inflammation and cancer progression. Through bioinformatics analysis, ATR, MCM2, and RFC4 are identified as potential therapeutic targets, and miRNA and transcription factor networks are explored to provide further insight into gene regulation. While the research presents valuable findings, the manuscript would benefit from improved clarity, particularly in the explanation of bioinformatics methodologies and the interpretation of immune infiltration results. A more structured discussion of the clinical implications of the findings and potential therapeutic strategies could enhance the manuscript’s impact. Overall, the study offers significant insights into the molecular links between LUAD and T2DM, contributing to a better understanding of their shared pathophysiology and potential treatment targets. Strengthening the methodological descriptions and refining the discussion would improve the accessibility and readability of the manuscript.
However, to strengthen the data and ensure clarity, the suggested revisions should be fully addressed.
Major revisions
Other observations/concerns
I strongly suggest that the authors standardize the manuscript by adding a space before citations. For example, in line 45: “… cause of cancer-related deaths worldwide [1].” instead of “… cause of cancer-related deaths worldwide[1].”
In the Materials and Methods section: software programs, servers, and packages – Including ssGSEA and CytoHubba – are mentioned without bibliographic references. I strongly suggest adding references (articles or the official website addresses used in the methods).
“ATR, MCM2, RFC4, NUP155, NUP107, NUP85, and NUP37” to “ATR, MCM2, RFC4, NUP155, NUP107, NUP85, and NUP37”. Gene names are often written in italics to distinguish them from other elements. Proteins encoded by genes, on the other hand, are not written in italics. This rule was established to facilitate the differentiation between genes (genomic elements) and gene products (proteins). Therefore, I strongly recommend that authors format all gene abbreviations used in the manuscript in italics.
Minor revisions
Line 44:
Rewrite the sentence for:
“... most common malignant ...” to “... most prevalent malignant ...”
Lines 45 and 46:
Rewrite the sentence for:
“The incidence and mortality of lung cancer are gradually increasing each year [2], ...” to “The incidence and mortality rates of lung cancer have been steadily increasing each year [2], ...”. In this way, the text is presented in a more fluid way.
Lines 48 and 49:
Rewrite the sentence for:
“It is noteworthy that Japan ...” to “Notably, Japan ...”
Line 50:
Similar to “non-small cell lung cancer (NSCLC)”, I suggest adding the acronym “SCLC” to “small cell lung cancer”.
Lines 51 and 52:
Rewrite the sentence for:
“Among NSCLC, lung adencarcinoma (LUAD) is the most common histological type.” to “Among NSCLC subtypes, lung adenocarcinoma (LUAD) is the most common histological variant.” In this way, the text is presented in a more fluid way.
Lines 52 to 54:
Rewrite the sentence for:
“Currently, surgical resection is applicable for clinical treatment of early-stage LUAD patients but not suitable for late-stage LUAD patients.” to “Currently, surgical resection is a viable treatment option for early-stage LUAD patients but is not suitable for those with late-stage disease.” In this way, the text is presented in a more fluid way.
Lines 58 to 59:
Rewrite the sentence for:
“… characterized by abnormal elevation of blood ...” to “… characterized by chronically elevated blood ...”
Lines 61 to 64:
Although the concepts are correct, the text would benefit from a broader and more comprehensive presentation: DM1 (Type 1 Diabetes mellitus) is caused by insufficient insulin secretion due to an autoimmune process that leads to the destruction of insulin-producing pancreatic β-cells. This results in an absolute insulin deficiency, making patients dependent on insulin therapy for glycemic control. On the other hand, DM2 (Type 2 Diabetes mellitus) is characterized by insulin resistance in target tissues (such as the liver, muscles, and adipose tissue) and the progressive dysfunction of pancreatic β cells, leading to insufficient insulin production relative to the body's demands.
Lines 64 to 66:
Rewrite the sentence for:
“Studies have found that diabetes is not only associated with microvascular complications such as nephropathy [9] and retinopathy [10], but also with cancers such as lung cancer, breast cancer, and pancreatic cancer [11], ...” to “Studies have shown that diabetes is associated not only with microvascular complications, such as nephropathy [9] and retinopathy [10], but also with an increased risk of cancers, including lung, breast, and pancreatic cancer [11], ...”. In this way, the text is presented in a more fluid way.
Line 106:
Rewrite the sentence for:
“"... while for GSE25724 (T2DM) and GSE10072 (LUAD) was GPL96.” to “... whereas GPL96 was used for GSE25724 (T2DM) and GSE10072 (LUAD).”
Line 108:
Rewrite the sentence for:
“The R software (version 4.1.3) packages “limma” and “GEOquery” ...” to “The R packages “limma” and “GEOquery” (version 4.1.3) ...”
Lines 110 and 111:
Rewrite the sentence for:
“DEGs were identified based on the criteria of adjusted P-value < 0.05 and |log2 fold change| > 1.5.” to “DEGs were identified using an adjusted p-value threshold of < 0.05 and an absolute log2 fold change > 1.5.”
Lines 120 and 122:
Rewrite the sentence for:
“The ssGSEA score was used to quantify the infiltration of immune cells in LUAD or T2DM tissues and determine the level of immune infiltration in each dataset sample.” to “The ssGSEA score was used to quantify immune cell infiltration in LUAD and T2DM tissues, providing an infiltration level for each dataset sample.” In this way, the text is presented in a more fluid way.
Lines 139 and 140:
Rewrite the sentence for:
“…network, and four algorithms, MNC, MCC, EPC, and Degree, were applied to confirm the final hub genes.” to “… network, applying four algorithms – MNC, MCC, EPC, and Degree – to confirm the key hub genes.”
Line 158:
Rewrite the sentence for:
“… criteria of p-value <0.05 and |log2 fold change|>1.5, …” to “… criteria of p-value < 0.05 and |log2 fold change| >1.5, ...”
Line 159:
Rewrite the sentence for:
“total of 4598 DEGs from GSE40791 and 2790 DEGs …” to “total of 4,598 DEGs from GSE40791 and 2,790 DEGs …”
Line 171:
Rewrite the sentence for:
“…we performed ssGSEA.” to “… we performed the ssGSEA analysis.
Line 172:
Rewrite the sentence for:
“… with …” to “… and ...”
Line 178:
Rewrite the sentence for:
“As shown in Figure 2B, the T2DM was ...” to “As shown in Figure 2B, T2DM was ...”
Lines 188 and 189:
Rewrite the sentence for:
“These genes were mainly involved in biological processes related to the positive ...” to “These genes were primarily involved in biological processes such as the positive ...”
Line 194:
Rewrite the sentence for:
“… associated with cytosol, …” to “… associated with the cytosol, …”
Lines 196:
Rewrite the sentence for:
“... involvement in transforming ...” to “... involvement in the transforming ...”
Line 203:
On line 203, adjust formation (start of paragraph).
Line 129:
Rewrite the sentence for:
“We constructed the utilized the miRNA-target gene network...” to “We constructed and utilized the miRNA-target gene network...”
Line 132
Rewrite the sentence for:
“… was…” to “…were…”
Lines 240 and 241:
Rewrite the sentence for:
“… and hsa-mir-1-3p by using miRPath4.0.” to “… and hsa-mir-1-3p using miRPath4.0.”
Lines 294 and 295:
Rewrite the sentence for:
“Therefore, the high glucose environment of T2DM may increase the activation of the TGF-β signaling pathway and the risk of developing LUAD.” to “Therefore, the hyperglycemic environment in T2DM may enhance TGF-β signaling activation, potentially increasing LUAD risk.”
Lines 354 to 356:
Rewrite the sentence for:
“In summary, this bioinformatics study identified the shared genetic characteristics between T2DM and LUAD, elucidating the potential molecular mechanisms underlying these two diseases.” to “In summary, this bioinformatics study identified shared genetic characteristics between T2DM and LUAD, shedding light on potential molecular mechanisms underlying their pathogenesis.”
Lines 354 to 360:
Rewrite the sentence for clarity:
“In summary, this bioinformatics study identified the shared genetic characteristics between T2DM and LUAD, elucidating the potential molecular mechanisms underlying these two diseases. We found that the imbalanced immune response and the TGF-β signaling pathway could be common pathogenic mechanisms in LUAD and T2DM. Additionally, ATR, MCM2, and RFC4 may serve as potential therapeutic targets for both diseases. Our findings may provide a certain basis for future treatment and prevention of T2DM and LUAD.” to “In summary, this bioinformatics study identified shared genetic characteristics between T2DM and LUAD, shedding light on potential molecular mechanisms underlying their pathogenesis. Our analysis suggests that dysregulated immune responses and aberrant TGF-β signaling may represent common pathogenic mechanisms linking LUAD and T2DM. Moreover, ATR, MCM2, and RFC4 emerge as potential therapeutic targets for both conditions, warranting further investigation. Our findings lay a foundation for future research aimed at developing novel therapeutic and preventive strategies for T2DM and LUAD.”
Author Response
- “The manuscript titled "Shared Genomic Features between Lung Adenocarcinoma and Type 2 Diabetes: A Bioinformatics Study" investigates the molecular and genetic connections between LUAD and T2DM, identifying shared differentially expressed genes (DEGs) and key pathways. The study highlights the role of immune dysregulation and the TGF-β signaling pathway in both diseases, emphasizing their contribution to inflammation and cancer progression. Through bioinformatics analysis,ATR, MCM2, and RFC4 are identified as potential therapeutic targets, and miRNA and transcription factor networks are explored to provide further insight into gene regulation. While the research presents valuable findings, the manuscript would benefit from improved clarity, particularly in the explanation of bioinformatics methodologies and the interpretation of immune infiltration results. A more structured discussion of the clinical implications of the findings and potential therapeutic strategies could enhance the manuscript’s impact. Overall, the study offers significant insights into the molecular links between LUAD and T2DM, contributing to a better understanding of their shared pathophysiology and potential treatment targets. Strengthening the methodological descriptions and refining the discussion would improve the accessibility and readability of the manuscript”
Response 1:Thank you for your comments. Per the Reviewer’s suggestions,We have carefully added a section to the discussion part of the paper, delving into an in - depth exploration of the clinical significance of the research findings and potential treatment strategies, aiming to provide more valuable references for research and practice in this field. (lines 391-411 )
- “I strongly suggest that the authors standardize the manuscript by adding a space before citations. For example, in line 45: “… cause of cancer-related deaths worldwide [1].” instead of “… cause of cancer-related deaths worldwide[1].”
Response 2:Thank you for your comments. Per the Reviewer’s suggestions, we have made revisions to the manuscript and incorporated the relevant content into the main text.(lines 45-100;295-390 )
- “software programs, servers, and packages – Including ssGSEA and CytoHubba – are mentioned without bibliographic references. I strongly suggest adding references (articles or the official website addresses used in the methods).”
Response 3:We thank the Reviewer for their valuable comments. Per the Reviewer’s suggestions, we have provided the official website addresses for the ssGSEA analysis and the cytoHubba plugin.(lines 139,160 )
- “ATR, MCM2, RFC4, NUP155, NUP107, NUP85, and NUP37”to “ATR, MCM2, RFC4, NUP155, NUP107, NUP85, and NUP37”. Gene names are often written in italics to distinguish them from other elements. Proteins encoded by genes, on the other hand, are not written in italics. This rule was established to facilitate the differentiation between genes (genomic elements) and gene products (proteins). Therefore, I strongly recommend that authors format all gene abbreviations used in the manuscript in italics.”
Response 4:We are truly grateful to the Reviewer for their insightful and valuable comments. Meanwhile, we sincerely apologize for any lack of clarity in our manuscript.Per the Reviewer’s suggestions, we have modified the font of gene names in the article.(lines 34-39,240-286,292,346-358,395-406 )
- “... most common malignant ...”to “... most prevalent malignant ...,The incidence and mortality of lung cancer are gradually increasing each year [2], ...” to “The incidence and mortality rates of lung cancer have been steadily increasing each year [2], ...,It is noteworthy that Japan ...” to “Notably, Japan ...,Similar to “non-small cell lung cancer (NSCLC)”, I suggest adding the acronym “SCLC” to “small cell lung cancer”,Among NSCLC, lung adencarcinoma (LUAD) is the most common histological type.” to “Among NSCLC subtypes, lung adenocarcinoma (LUAD) is the most common histological variant,Currently, surgical resection is applicable for clinical treatment of early-stage LUAD patients but not suitable for late-stage LUAD patients.” to “Currently, surgical resection is a viable treatment option for early-stage LUAD patients but is not suitable for those with late-stage disease,… characterized by abnormal elevation of blood ...” to “… characterized by chronically elevated blood ...,Although the concepts are correct, the text would benefit from a broader and more comprehensive presentation: DM1 (Type 1 Diabetes mellitus) is caused by insufficient insulin secretion due to an autoimmune process that leads to the destruction of insulin-producing pancreatic β-cells. This results in an absolute insulin deficiency, making patients dependent on insulin therapy for glycemic control. On the other hand, DM2 (Type 2 Diabetes mellitus) is characterized by insulin resistance in target tissues (such as the liver, muscles, and adipose tissue) and the progressive dysfunction of pancreatic β cells, leading to insufficient insulin production relative to the body's demands,Studies have found that diabetes is not only associated with microvascular complications such as nephropathy [9] and retinopathy [10], but also with cancers such as lung cancer, breast cancer, and pancreatic cancer [11], ...” to “Studies have shown that diabetes is associated not only with microvascular complications, such as nephropathy [9] and retinopathy [10], but also with an increased risk of cancers, including lung, breast, and pancreatic cancer [11],... while for GSE25724 (T2DM) and GSE10072 (LUAD) was GPL96.” to “... whereas GPL96 was used for GSE25724 (T2DM) and GSE10072 (LUAD),The R software (version 4.1.3) packages “limma” and “GEOquery” ...” to “The R packages “limma” and “GEOquery” (version 4.1.3) ...,DEGs were identified based on the criteria of adjusted P-value < 0.05 and |log2 fold change| > 1.5.” to “DEGs were identified using an adjusted p-value threshold of < 0.05 and an absolute log2 fold change > 1.5,The ssGSEA score was used to quantify the infiltration of immune cells in LUAD or T2DM tissues and determine the level of immune infiltration in each dataset sample.” to “The ssGSEA score was used to quantify immune cell infiltration in LUAD and T2DM tissues, providing an infiltration level for each dataset sample,…network, and four algorithms, MNC, MCC, EPC, and Degree, were applied to confirm the final hub genes.” to “… network, applying four algorithms – MNC, MCC, EPC, and Degree – to confirm the key hub genes,… criteria of p-value <0.05 and |log2 fold change|>1.5, …” to “… criteria of p-value < 0.05 and |log2 fold change| >1.5, ...,total of 4598 DEGs from GSE40791 and 2790 DEGs …” to “total of 4,598 DEGs from GSE40791 and 2,790 DEGs …,…we performed ssGSEA.” to “… we performed the ssGSEA analysis..........................In summary, this bioinformatics study identified the shared genetic characteristics between T2DM and LUAD, elucidating the potential molecular mechanisms underlying these two diseases. We found that the imbalanced immune response and the TGF-β signaling pathway could be common pathogenic mechanisms in LUAD and T2DM. Additionally, ATR, MCM2, and RFC4 may serve as potential therapeutic targets for both diseases. Our findings may provide a certain basis for future treatment and prevention of T2DM and LUAD.” to “In summary, this bioinformatics study identified shared genetic characteristics between T2DM and LUAD, shedding light on potential molecular mechanisms underlying their pathogenesis. Our analysis suggests that dysregulated immune responses and aberrant TGF-β signaling may represent common pathogenic mechanisms linking LUAD and T2DM. Moreover, ATR, MCM2, and RFC4 emerge as potential therapeutic targets for both conditions, warranting further investigation. Our findings lay a foundation for future research aimed at developing novel therapeutic and preventive strategies for T2DM and LUAD.”
Response 5:We are truly grateful to the Reviewer for their insightful and valuable comments. Meanwhile, we sincerely apologize for any lack of clarity in our manuscript.Based on the suggestions of the review experts, we have comprehensively and meticulously revised this part of the article, striving to make the content more in line with academic norms and requirements.(introduction page1,2; 139-141;161,183,197,204,215,222,260,272,329 )

Reviewer 2 Report
Comments and Suggestions for Authors
This is a very meaningful study, but the author is unclear in many places during the description, especially in the materials and methods section, which directly affects the reliability of the data analysis results. For example, what is the criteria for GEO data screening? Why are there only two in some places when a total of 4 are screened? This directly leads to unclear logical structure and inability to judge the reliability of the results. The author needs to make serious revisions.
Other minor issues but not limited to these:
In line 97, the author wrote that lung adenocarcinoma was used as a keyword to search the GEO database. What are the specific results? What are the screening criteria? Why did they directly select those four as candidates for analysis?
In line 106, will the differences in sequencing platforms affect the results? This needs to be explained.
In line 115, why were GSE10072 and GSE29221 selected as validation data sets? What are the specific selection criteria?
142-144 Why did the ROC analysis not use the other two data sets?
The words on Figure 4A are completely unclear and need to be revised?
Why are line 218 GSE10072 (LUAD) and GSE71416 (T2DM) not reflected?
The words in Figure 6 overlap each other and are not clear. They need to be revised.
The English could be improved to more clearly express the research.
Author Response
- “what is the criteria for GEO data screening? Why are there only two in some places when a total of 4 are screened?”
Response 1:We are truly grateful to the Reviewer for their insightful and valuable comments. Meanwhile, we sincerely apologize for any lack of clarity in our manuscript.Per the Reviewer’s suggestions, we have supplemented the screening criteria for GEO datasets in the Materials and Methods section of the article. In this study, we first used GSE40791 (LUAD) and GSE25724 (T2DM) to screen for differentially expressed genes, and then applied a series of methods to determine the core target genes. To further validate the stability and consistency of the core genes, we used the external validation datasets GSE10072 (LUAD) and GSE71416 (T2DM). The supplementary part has been completely added to the article materials. (lines 103-127)
- “In line 97, the author wrote that lung adenocarcinoma was used as a keyword to search the GEO database. What are the specific results? What are the screening criteria? Why did they directly select those four as candidates for analysis?”
Response 2:We are truly grateful to the Reviewer for their insightful and valuable comments. Meanwhile, we sincerely apologize for any lack of clarity in our manuscript.Searching the GEO database with "lung adenocarcinoma" as the keyword, a series of gene expression datasets related to lung adenocarcinoma were obtained. These datasets cover data from different patient groups, such as tumor tissue samples and normal tissue samples. There are differences among the datasets in terms of sample size, sample source, experimental methods, and the types and quantities of genes detected.During screening, datasets with a high proportion of missing data, irregular sample annotations, and insufficient sample sizes were excluded. Finally, these four datasets were selected as candidates. This is because, compared with other datasets, their data are more complete, the sample annotations are in line with the norms, and the sample sizes can fully meet the requirements of this study, which helps to ensure the accuracy and reliability of the subsequent analysis results.
- “In line 106, will the differences in sequencing platforms affect the results? This needs to be explained.”
Response 3:We are truly grateful to the Reviewer for their insightful and valuable comments. Meanwhile, we sincerely apologize for any lack of clarity in our manuscript.The differences in sequencing platforms can affect the results. Different GPL platforms have different probe designs and coverage. If the probe specificity is poor, it will affect the data accuracy. If the coverage is different, the information in some regions cannot be detected. GPL570 can detect 54,000 transcripts, which can comprehensively present the gene expression situation, facilitate the discovery of new disease - related genes and biological pathways, be applicable to a variety of biological research, and provide a comprehensive basis for disease diagnosis, treatment, and drug development. GPL96 targets about 14,500 known important genes, can meet the needs of expression analysis of common key genes, and the analysis results can be obtained quickly, which is conducive to preliminary exploration and rapid screening of key genes.
- “In line 115, why were GSE10072 and GSE29221 selected as validation data sets? What are the specific selection criteria?”
Response 4:We are truly grateful to the Reviewer for their insightful and valuable comments. Meanwhile, we sincerely apologize for any lack of clarity in our manuscript.The external validation datasets we selected are GSE10072 (LUAD) and GSE71416 (T2DM) respectively. Regarding the screening criteria for these datasets, we have detailedly supplemented and improved them in the Methods section of the article. We are very sorry that the content was not clear due to writing errors, which have caused you trouble in reading.
- “142-144 Why did the ROC analysis not use the other two data sets?”
Response 5:We are truly grateful to the Reviewer for their insightful and valuable comments. Meanwhile, we sincerely apologize for any lack of clarity in our manuscript.First, we utilized the training datasets GSE40791 and GSE25724 to validate the seven central genes and calculated the areas under the receiver operating characteristic (ROC) curves. The results showed that the areas under the curves of these seven genes were all greater than 0.7, indicating that they possess good predictive abilities. Subsequently, to further verify the reliability and universality of the predictive performance of these seven genes, we employed the external validation datasets GSE10072 and GSE71416 to conduct an in - depth validation of their predictability.
- “The words on Figure 4A are completely unclear and need to be revised?”
Response 6:We are truly grateful to the Reviewer for their insightful and valuable comments. Meanwhile, we sincerely apologize for any lack of clarity in our manuscript.Per the Reviewer’s suggestions,We have made modifications to Figure 4A in the article.
- “Why are line 218 GSE10072 (LUAD) and GSE71416 (T2DM) not reflected?”
Response 7:We are truly grateful to the Reviewer for their insightful and valuable comments. Meanwhile, we sincerely apologize for any lack of clarity in our manuscript.Given that GSE10072 and GSE71416 are external validation datasets, they are only applied in the research to validate the predictive performance of the area under the receiver operating characteristic (ROC) curve of the central genes.
- “The words in Figure 6 overlap each other and are not clear. They need to be revised.”
Response 8:We are truly grateful to the Reviewer for their insightful and valuable comments. Meanwhile, we sincerely apologize for any lack of clarity in our manuscript.Per the Reviewer’s suggestions,We have made modifications to Figure 6 in the article.

Round 2
Reviewer 2 Report
Comments and Suggestions for Authors
The author has made the revisions as requested, but the language needs further improvement.
Comments on the Quality of English LanguageThe English could be improved to more clearly express the research.
Author Response
Reviewer 1 Comment 1: The authors have made the revisions as requested, but the language needs further improvement.
Response 1: We really appreciate you taking the time to review our article and provide very constructive suggestions. These suggestions are of great help in improving the quality of the article. We would like to express our sincerest gratitude to you. Based on your suggestions, we have thoroughly polished the text. Thank you again for your careful guidance. We look forward to your further feedback.
